# Haemostasis and Inflammatory Parameters as Potential Diagnostic Biomarkers for VTE in Trauma-Immobilized Patients

**DOI:** 10.3390/diagnostics13010150

**Published:** 2023-01-02

**Authors:** Noor Nabila Ramli, Salfarina Iberahim, Noor Haslina Mohd Noor, Zefarina Zulkafli, Tengku Muzaffar Tengku Md Shihabuddin, Mohd Hadizie Din, Muhamad Aizat Mohamed Saat, Ahmad Hadif Zaidin Samsudin

**Affiliations:** 1Hospital Universiti Sains Malaysia, Universiti Sains Malaysia, Kubang Kerian 16150, Malaysia; 2Department of Hematology, School of Medical Sciences, Universiti Sains Malaysia (USM), Kubang Kerian 16150, Malaysia; 3Department of Orthopedic, School of Medical Sciences, Universiti Sains Malaysia (USM), Kubang Kerian 16510, Malaysia; 4Department of Radiology, School of Medical Sciences, Universiti Sains Malaysia (USM), Kubang Kerian 16150, Malaysia

**Keywords:** biomarkers, trauma, immobilized, deep venous thrombosis, pulmonary embolism, venous thromboembolism

## Abstract

Venous thromboembolism (VTE), which encompasses deep venous thrombosis (DVT) and pulmonary embolism (PE), is a major public health concern due to its high incidences of morbidity and mortality. Patients who have experienced trauma with prolonged immobilization are at an increased risk of developing VTE. Plasma D-dimer levels have been known to be elevated in trauma patients, and they were closely correlated with the number of fractures. In other words, plasma D-dimer levels cannot be used as the only indicator of VTE in trauma cases. Given the limitations, further study is needed to explore other potential biomarkers for diagnosing VTE. To date, various established and novel VTE biomarkers have been studied in terms of their potential for predicting VTE, diagnostic performance, and improving clinical therapy for VTE. Therefore, this review aims to provide information regarding classic and essential haemostasis (including prothrombin time (PT), activated partial thromboplastin time (aPTT), D-dimer, fibrinogen, thrombin generation, protein C, protein S, antithrombin, tissue factor pathway inhibitor, and platelet count) and inflammatory biomarkers (C-reactive protein, erythrocyte sedimentation rate, and soluble P-selectin) as potential diagnostic biomarkers that can predict the risk of VTE development among trauma patients with prolonged immobilization. Thus, further advancement in risk stratification using these biomarkers would allow for a better diagnosis of patients with VTE, especially in areas with limited resources.

## 1. Introduction

Venous thromboembolism (VTE) includes two interrelated conditions, deep venous thrombosis (DVT) and pulmonary embolism (PE). It is the third most common vascular disease after myocardial infarction and stroke. VTE is a complex disorder that involves interactions between clinical risk factors and a thrombotic predisposition that may be acquired or inherited. An estimated 300,000 to 600,000 people in the United States develop DVT and/or PE each year, with one third of patients presenting with PE, which often result in sudden death [1].

Several major risk factors predispose patients to thromboembolic complications, including cancer, stroke, prolonged immobilization, increasing age, hormonal treatment, pregnancy or puerperium, extended air travel, congestive heart failure, acute inflammatory bowel disease, and nephrotic syndrome [2,3,4]. These thromboembolic events contribute significantly to mortality and morbidity, and treating these diseases will impose a high burden on the healthcare system [5].

Despite the extensive use of preventative measures for trauma patients, incidences of VTE remain high after an injury. In trauma patients, VTE is the most frequent and avoidable cause of in-hospital deaths [2,6]. Thus, it would be significantly advantageous to have biomarkers that allow early identification of trauma patients who are at risk of developing the disorder, so that VTE prevention strategies can be implemented. In this review, we will focus on investigating potential diagnostic biomarkers for the prediction of VTE in trauma-immobilized patients, which include haemostasis and inflammatory parameters.

## 2. VTE in Trauma and Immobilized Patients

Trauma is one of the triggers of hypercoagulable states, which may complicate patients’ conditions [7] and lead to poor clinical outcomes. Multiple studies have found that the incidence rate of DVT in trauma patients without prophylaxis varies from 5% to 80% [8,9,10], with head injury, lower limb injury, pelvic fracture, and prolonged immobilization having traditionally been regarded as a risk factor for VTE [10]. The reported prevalence of DVT following trauma varies depending on numerous factors, including patient demographics, the nature, site and degree of injury, method of detection for VTE, and mode of prophylaxis, resulting in a broad spectrum of numbers in the literature [9,11]. Given the possibility of poor outcomes for VTE patients, as well as the potential for bleeding associated with anti-coagulants, VTE should be correctly diagnosed and managed when present, and it should be securely excluded when absent [12].

The mechanism by which venous thrombi develop is classically explained by the presence of three predisposing factors known as the Virchow’s triad (Figure 1). The Virchow’s triad is made up of three major components: abnormal blood flow (venous stasis), vascular endothelial injury (vessel injury), and changes in the clotting mechanism (hypercoagulability). Major trauma often triggers one or more risk elements in the Virchow’s triad, leading to thrombosis.

Trauma patients suffering from spinal cord injuries, head injuries, pelvic fractures, and long bone fractures are frequently immobilized. This immobilization results in a static posture and may induce venous stasis. Blood/venous stasis leads to thrombosis. At the same time, the direct trauma toward the vessel walls will expose the blood to the von Willebrand factor, tissue factor, and collagen. These substances will attract platelets and promote both intrinsic and extrinsic coagulation cascades, resulting in hypercoagulability and, subsequently, thrombosis [13]. A few studies have reported that natural anticoagulants were reduced after major trauma [14,15,16], and these conditions may induce a hypercoagulable state in patients.

Immobilization is defined as the restriction or elimination of the movement of a body component by mechanical methods or complete bed rest to enable recovery. The Wells score describes immobility as a reduced motion for three days or longer. One of the risk factors of VTE is immobility for more than 72 h. The development of the Wells score established a clinical prediction rule to direct further diagnostic testing. However, the score was intended for non-traumatic conditions rather than trauma patients. In addition, it was unable to predict PE in patients hospitalized for orthopaedic trauma care [17]. 

A hypercoagulable state combined with prolonged immobility increases the risk of VTE. These issues have been widely recognized as a complicated component in managing trauma patients during and after hospitalization. Given the high morbidity and mortality risk associated with VTE, preventing, detecting, and treating these problems is crucial for trauma patients [13].

A review of six randomized controlled trials found that the incidence rates of DVT vary from 4.3% to 40% in patients who had cast immobilization for at least one week without receiving prophylaxis [18]. Another study supported these findings and showed that below-knee cast immobilization significantly increased the risk of VTE [19]. 

A retrospective study by Yumoto et al. involving 204 severely injured trauma patients found that they had potential VTE development at a median of 10 days following admission to intensive care units [20]. Brakenridge et al. supported these findings, where they also observed that the average time to develop VTE is roughly 10 days following injury [21]. In a MEGA study, which studies VTE’s etiology, researchers discovered a significant correlation between the duration of immobility and VTE development. Twice as many patients were diagnosed with VTE in the second week of immobilization than in the first week, which correlated with the natural course of the disease as blood clots take time to develop. It was also shown that patients in hypercoagulable states generated by trauma-induced immobilization had a higher prevalence of VTE than those who were immobilized without trauma or tissue injury [19].

Selby et al., in their study on haemostatic changes and their relationship to VTE, found that the clinical risk factors, which are age (in increments of one year), gender, and mobility scores, were independent predictors of VTE following major trauma [7]. They used a five-point scale for the mobility score: 1. Bedridden with only passive leg movements; 2. Bedridden with active leg movements; 3. Mobile with the use of a wheelchair but not weight bearing; 4. Mobile with partial weight bearing or with aid; and, 5. Fully mobile without aid. They found that patients who had DVT had low mobility scores [8].

## 3. Haemostasis and Inflammatory Parameters as Potential Diagnostic Biomarkers in Diagnosing VTE

A biological marker, also known as a biomarker, is a measured indicator that corresponds well with a disease’s risk or development. It could be a gene, an enzyme, a tissue, or blood. Biomarkers could also be a general characteristic alteration in a biological structure/component or an alteration in a complex organ function. Numerous studies have discovered that haemostasis and inflammatory biomarkers are associated with a higher risk of VTE development. D-dimer level is an example of a haemostatic biomarker that has been effectively utilized in clinical practice [22]. A retrospective analysis by Ma et al. investigating the association between haematological biomarkers and DVT found that D-dimer level >0.5 mg/L and platelet distribution width <12% were associated with DVT [23]. However, plasma D-dimer level was known to be elevated in trauma patients, and it was closely correlated with the number of fractures. In other words, it cannot be used as an indicator of VTE in trauma cases [24]. Given its limitation, further study is needed to explore other potential biomarkers for diagnosing VTE in a limited setting. Other haemostasis and inflammatory parameters will be further discussed in the following sections.

### 3.1. Effect of Trauma and Immobilization on Haemostasis Parameters

Haemostasis is a complex and well-balanced physiologic mechanism that maintains blood fluid. Following a traumatic vascular rupture, haemostatic proteins encourage clot formation to limit excessive bleeding. Meanwhile, other haemostatic proteins will act as an anticoagulant, lysing the clot and restricting its extension. The three consecutive stages of haemostasis are primary haemostasis, secondary haemostasis, and tertiary haemostasis. Primary haemostasis describes the actions of platelets. Secondary haemostasis is the formation of a stabilized fibrin clot through the coagulation cascade. Tertiary haemostasis comprises the fibrinolytic and anticoagulant systems, which slow down the coagulation cascade, regulate runaway coagulation, and disintegrate a clot when it is no longer required [25]. Therefore, any imbalance in this system will lead to complications.

#### 3.1.1. Prothrombin Time (PT) and activated Partial Thromboplastin Time (aPTT)

The PT and aPTT are global coagulation screening tests used to determine the coagulation status in people who have acquired deficiencies in coagulation factors from the intrinsic and extrinsic coagulation pathways [26]. The PT test measures the time required for patients’ plasma to clot after the addition of calcium and thromboplastin, while the aPTT test represents the time for patients’ plasma to clot following the addition of phospholipid and calcium [27]. The PT test is sensitive to factors II, V, VII, X, and fibrinogen deficiency, whereas the aPTT is sensitive to factors II, V, VIII, IX, X, XI, XII, and fibrinogen deficiency [26]. 

According to Yuan et al., the PT test shows higher sensitivity than the aPTT test to low coagulation factor levels in acute trauma patients, with a sensitivity of 84% and 50%, respectively. Hence, the PT cut-off value is a more reliable indicator for detecting an underlying coagulation factor deficiency in patients [28]. In contrast, the Atherosclerosis Risk in Communities (ARIC) study cohort by Zakai et al. reported that lower aPTT was associated with a two-fold increased risk of VTE in the 13 years of follow-up [29]. Lower aPTT levels may be associated with a higher risk of thrombosis due to increased activity of coagulation factors in intrinsic or common pathways [30].

A recent study by Cao et al. observed a significant increase in PT levels in trauma patients with DVT (12.63 ± 0.89 s) compared with trauma non-DVT patients (12.23 ± 0.76 s) and healthy control (12.00, IQR = 0.70 s). On the other hand, the aPTT levels in trauma patients with DVT (24.76 ± 3.03 s) were significantly decreased compared with trauma non-DVT patients (25.28 ± 3.51 s) and healthy control (27.07 ± 2.24 s). They also reported that the optimal cut-off value for PT in diagnosing DVT was 12.05 s with a sensitivity of 72.92% and specificity of 47.92% (Area under the curve, or AUC, 0.617, 95% CI 0.505–0.730, *p* = 0.048) [31].

According to Park et al., the hypercoagulable state is frequently underestimated by commonly used diagnostic tests, such as PT and aPTT. However, it could be identified through the use of thromboelastography (TEG). Only 6% of individuals with prolonged PT and aPTT but low protein C and AT were found to have a pulmonary embolism. Even though the TEG test is better than the PT and aPTT tests at detecting hypercoagulability states in trauma patients, the TEG test could not be used to distinguish between patients with VTE complications and those without [32].

#### 3.1.2. D-dimer

D-dimer forms after thrombin-generated fibrin clots are broken down by plasmin, indicating general stimulation of blood coagulation and fibrinolysis. It is a cross-linked fibrin degradation product, and its level is one of the screening tests used to evaluate the hypercoagulability state and the risk of VTE [33]. This screening test has been utilized as a tool in many studies before proceeding to diagnostic tests, such as Doppler ultrasonography for DVT and CT pulmonary angiography for PE. It is used in the emergency room as an initial screening test to aid in diagnosing VTE. According to studies by Vanfleteren and Wesseling, the measurement of D-dimer levels is a valuable test for suspected VTE patients in primary care settings [34]. 

Cao et al., in a study, found that the level of D-dimer was significantly higher in the trauma DVT group (7.30, IQR = 5.85) compared with the trauma non-DVT group (3.31, IQR = 10.68). They also observed that the optimal cut-off value for D-dimer level was 3.825 mg/L, with a sensitivity of 85.42% and specificity of 51.11% (AUC 0.624, 95% CI 0.505–0.744, *p* = 0.039) for diagnosing DVT in trauma patients [31]. Another research found that a preoperative D-dimer value that is higher than 4.01 mg/L was a significant predictor of preoperative DVT in traumatic fracture patients, with a preoperative D-dimer AUC of 0.593, 71.2% sensitivity and 44.8% specificity [35]. 

Cheng et al. revealed that levels of D-dimer, fibrinogen and plasminogen activator inhibitor-1 (PAI-1) were significantly increased in DVT patients after a lower limb fracture surgery. They also demonstrated that the AUC of D-dimer, fibrinogen, PAI-1 and the combination of these three biomarkers for postoperative DVT in lower limb fracture patients was 0.966, 0.429, 0.792, and 0.992, respectively. As a result, early monitoring of D-dimer, fibrinogen, and PAI-1 levels is a strong predictor of postoperative thrombosis [36]. In another study on post-operative DVT after total knee arthroplasty (TKA), researchers found that patients with postoperative DVT had a high level of D-dimer on days 1 and 7, but in cases with no underlying DVT, it returned to normal levels [37].

The D-dimer test is an effective, non-invasive triage test with a strong predictive value in individuals with suspected VTE. A negative D-dimer level result, combined with a low pre-test clinical probability of disease using the Wells DVT prediction method, may safely rule out VTE [38] and reduce the number of patients who require additional imaging techniques [39]. As mentioned earlier, D-dimer level is commonly increased in post-trauma patients, and the plasma D-dimer level was correlated with injury severity. Given its limitation, it cannot be used as the only marker to predict VTE in trauma patients. Other potential biomarkers are needed to predict VTE in trauma patients and, at the same time, reduce unnecessary radiological examinations. 

#### 3.1.3. Fibrinogen

Fibrinogen is a soluble glycoprotein found in the plasma. It is synthesized in the liver. Plasma fibrinogen is an essential component of the coagulation cascade and a significant predictor of blood viscosity and blood flow [40]. In the case of trauma, the concentration of this protein in the blood may rise fourfold above the baseline. A few studies found that an increase in plasma fibrinogen levels was associated with an increased risk of hypercoagulable states, which lead to VTE. According to one study in mice, hyperfibrinogenemia will promote thrombosis and resist thrombolysis [41].

Fibrinogen’s role in trauma or inflammation is often defined as proinflammatory. It was discovered that mice with fibrinogen deficiency had a delayed inflammatory response to intravenous endotoxin, suggesting that physiologic levels of fibrinogen contributed to the commencement of inflammation [42]. 

Fibrinogen levels were found to be significantly increased in the trauma DVT group (362.70 ± 117.83 mg/dL) compared with the trauma non-DVT group (358.05 ± 152.88 mg/dL) and healthy control (247.90 ± 43.08 mg/dL) [31]. In a retrospective study analyzing various data (D-dimer, fibrinogen, C-reactive protein (CRP), ultrasound, and others) in traumatic fracture patients, researchers reported that fibrinogen tests were potentially useful in predicting postoperative VTE. The ROC analysis of this study shows that the AUC for fibrinogen was 0.5209, with a cut-off point value of 3.543 g/L for the diagnosis of VTE in elderly fracture patients. Furthermore, it was also found that a combination of fibrinogen and D-dimer value can effectively diagnose VTE in traumatic fracture patients older than 60 years old with an accuracy of 0.729 [43]. Therefore, fibrinogen may be a promising biomarker that can potentially predict sub-clinical and postoperative VTE.

#### 3.1.4. Thrombin Generation

Thrombin is a vital enzyme in the coagulation process that converts fibrinogen to fibrin, resulting in the production of clots. Although thrombin production is not a classical biomarker, it may be detected in plasma using a chromogenic or fluorescent substrate and recorded in a thrombin generation (TG) curve. Several characteristics indicating thrombin activity may be derived from the graph, including the lag time (the time between thrombin bursts), the peak value of thrombin (the maximal concentration of thrombin formed at a given point in time), and the AUC (endogenous thrombin potential), which represents the total amount of thrombin generated [44,45]. The thrombin generation assay (TGA) is a worldwide test that records the combined effects of thrombin generation and thrombin inactivation in real time. TGA, therefore, represents the outcome of procoagulant and anticoagulant activities in the blood and plasma [46].

It has been shown that TG is a risk factor for VTE [47] and may be used as a prediction marker for assessing thrombosis [48]. Several research has also looked into the relationship between high TG and the probability of having a first and recurring VTE episode. Endogenous thrombin potential and peak thrombin levels were shown to be higher in patients with a history of VTE compared with healthy persons, and they were linked to the likelihood of a new VTE incident [49,50]. 

Park et al., in their study of VTE biomarkers following acute trauma, found that increasing age, body mass index (BMI) ≥ 30 kg/m^2^, any surgery requiring general anesthesia, and first-time to peak (ttpeak) value (2.26 [1.61, 3.18] per 1 (minute) decrease, *p* < 0.0001) appear to be independent predictors of VTE within 92 days following trauma in a multivariable analysis. They also demonstrated that the addition of the first ttpeak to another independent predictor (age, BMI, any surgery) will accurately predict the occurrence of VTE following injury and reduce the rate of misclassification by 8.7% (10/115) [51]. In a pilot study involving 64 trauma patients, they observed a significant increase in lag time and ttpeak in trauma patients compared with the healthy control for the whole blood assay. Nonetheless, in the plasma thrombin generation assay, there was no difference in terms of lag time between trauma patients and healthy control, while there was a slight decrease in ttpeak in trauma patients. Both endogenous thrombin potentials show an increase in trauma patients compared with healthy control in the plasma and whole blood thrombin generation parameters [52]. 

#### 3.1.5. Protein C, Protein S, and Antithrombin

Several studies have shown that following trauma, thrombin generation markers [7] and tissue factors were increased [53]. In contrast, natural anticoagulants, such as protein C (PC) [14], protein S (PS) [16], and antithrombin (AT), decreased [15]. Trauma to a blood vessel triggers the production of procoagulant substances, which may lead to platelet-leukocyte adhesion and aggregation. Endothelial dysfunction caused by trauma-induced tissue factor-bearing microparticles may promote thrombin generation and another procoagulant process that favors pathological thrombosis [54,55].

PC and PS are members of the vitamin K-dependent glycoprotein family. Thrombin activates PC to become activated protein C (APC), and once APC is formed, it binds to PS, a non-enzymatic cofactor, on the surface of activated cells. This combination will then inactivate factors Va and VIIIa by restricting proteolysis. APC will inhibit thrombin generation by deactivating the active clotting factors Va and VIIIa (Figure 2). Sixty percent of PS is attached to the C4b-binding protein, while 40% circulates as free PS [56], and only free PS is functionally active [57]. Inherited PC, PS, and AT deficiencies have been associated with a higher risk of recurrent thrombosis.

Owing et al. examined 157 individuals with severely injured post-trauma and discovered that 61% had reduced AT levels [15]. Engelman et al., meanwhile, reported that functional PC was reduced in trauma patients with more severe injuries [14]. 

#### 3.1.6. Platelet

Platelets and coagulation factors play a major role in haemostasis in trauma and injuries. Platelets are megakaryocyte cytoplasm fragments that are produced from the bone marrow and hence have no nucleus. This blood component will be involved in primary haemostasis and followed by secondary haemostasis in the case of trauma or injury. The average platelet count range is between 150 to 400 × 10^9^/L. Thrombocytosis occurs when the platelet count is beyond the range [58].

Thrombocytosis can be classified into two categories: primary and secondary thrombocytosis. Primary thrombocytosis is often caused by clonal problems, such as myeloproliferative neoplasm, whereas secondary thrombocytosis includes reactive events, such as acute bleeding, infection, and inflammation [59]. Trauma can induce secondary thrombocytosis. According to Valade et al., from a total of 176 trauma patients examined, 20.4% developed thrombocytosis. The disorder is one of the factors of thromboembolic events when combined with other clinical risk factors [60].

The pathogenesis of this complication develops as a result of the body’s reaction to trauma or injury. In response to trauma, the body produces more cytokines, such as interleukin-6 (IL-6). IL-6 can induce thrombocytosis through its action on thrombopoietin [61]. Thrombopoietin is the ligand of the c-mpl proto-oncogene. It also functions as the primary regulator and promoter of megakaryocyte progenitor proliferation and differentiation [62]. In a 2022 study, Zhang et al. identified that there was a significant difference in the platelet counts (*p* < 0.001) between the thrombus group (176 × 10^9^/L) and the non-thrombus group (142 × 10^9^/L) among elderly patients with hip fractures. According to the multiple logistic regression analysis of this study, the platelet count was one of the independent risk factors of DVT (*p* < 0.05), with an AUC = 0.642 (95% CI: 0.569~0.714), the cutoff value of 200.5 × 10^9^/L, and sensitivity and specificity of 38.9% and 85.9%, respectively [63].

#### 3.1.7. Tissue Factor Pathway Inhibitor

Tissue factor pathway inhibitors (TFPI) are a natural anticoagulant protein in the extrinsic coagulation pathway. TFPI are produced primarily by endothelial cells and found within the vessel wall for about 80%, whereas only 20% circulates in the plasma [64,65]. The TFPI concentration in plasma is very low, with the majority (80%) being bound to lipoproteins, while the remaining 20%, known as free TFPI, are unbound and thus have the primary anticoagulant effect [66]. The anticoagulant effect of TFPI can be divided into two processes. First, they interact with activated FXa to produce the TFPI/FXa complex, which then interacts with the FVIIa/TF complex through the TFPI to form an inactive quaternary complex (TF/VIIa/Xa/TFPI) [67,68]. 

In a prospective case-control study in 2021, Cao et al. investigated the role of TFPI in the trauma group (with and without DVT) and healthy control. They reported that the anticoagulant activity of TFPI (the levels of TFPI initial anticoagulant time ratio, TFPI whole anticoagulant time ratio, and TFPI anticoagulant rate) in trauma patients with DVT (39.91 ± 13.44%; 13.65 ± 7.96%; and 32.61 ± 15.43%) were significantly higher than those in trauma patients without DVT (32.19 ± 10.28%; 9.77 ± 4.91%; and 25.87 ± 10.32%). They also found that the optimal cutoff value for the TFPI initial anticoagulant time ratio for diagnosing DVT in trauma patients was 33.53% with a sensitivity of 71.11% and specificity of 55.32% (AUC 0.646, 95% CI 0.533–0.759) [31]. A similar result was observed by Sidelman et al. in a study on acute DVT patients [69]. The elevated levels of TFPI anticoagulant activity in trauma patients may have been due to severe injury of vascular endothelial cells and inflammation induced by traumatic fractures [70]. Furthermore, an increased in TFPI levels may also represent a stress response to the hypercoagulable state in trauma patients with DVT [31]. Thus, TFPI parameters can potentially be used as a biomarker for predicting DVT, particularly in trauma patients. Interestingly, another study shows a contradicting result, where researchers observed that a reduction in the TFPI concentration was a risk factor for DVT [71].

### 3.2. Effect of Trauma and Immobilization on Inflammatory Parameters

Trauma or an injury can initiate an acute phase response which consists of local and systemic responses. Local reactions include vasodilation with the release of lysosomal enzymes and platelet aggregation. Meanwhile, systemic reactions include increased acute phase protein, fever, and leukocytosis. CRP and erythrocyte sedimentation rate (ESR) are acute phase markers that reflect acute phase responses [72].

#### 3.2.1. C- Reactive Protein (CRP)

CRP was discovered by Tillet and Francis in 1930. It was detected in the serum of pneumonia patients. CRP is synthesized by hepatocytes. CRP plasma levels are generally 1 mg/L in healthy adults, with normal values defined as less than 10 mg/L. These plasma levels of CRP will rise after four to six hours of acute tissue damage and will keep rising several folds within 24 to 48 h. Once the tissue structure is rebuilt, the CRP levels will return to normal [72]. The connection between thrombosis and inflammation in the development of venous thrombosis was validated by studies that observed an increase in plasma CRP levels in individuals with DVT compared with controls (2.5 mg/dL ± 3.2 vs. 1.0 mg/dL ± 2.5, *p* < 0.001) [73]. 

#### 3.2.2. Erythrocyte Sedimentation Rate (ESR)

ESR was proposed by Westergren in 1921. Westergren’s technique measures the rate of gravity settling of anticoagulated red blood cells in one hour from a fixed location in a calibrated tube of a certain length and diameter kept upright [74]. Erythrocytes often contain net negative charges and hence resist one other. However, following trauma, high molecular weight proteins that are positively charged, such as fibrinogen, would have increased, favoring rouleaux formation and, as a result, increase the ESR. Based on this result, ESR was utilized as an indirect measure of the acute phase response [72].

The ESR value may be influenced by red blood cell size and shape, fluid state, and plasma composition. Moreover, a smoking habit, temperature, and medicines, such as nonsteroidal anti-inflammatory drugs (NSAIDs), all have an effect on ESR. In a retrospective study involving 584 knee osteoarthritis patients undergoing TKA, Xiong and Cheng reported that the increase in ESR values was associated with the development of DVT before TKA [75]. Although the finding was based on the examination of knee osteoarthritis patients, the relationship between ESR and VTE may have a similar value to the raised ESR in trauma-immobilized patients. However, this association requires further study.

#### 3.2.3. Soluble P-Selectin

P-selectin is an adhesion glycoprotein that is present in the α-granules of platelets and the Weibel–Palade bodies of endothelial cells. P-selectin is translocated to the cell surface following the activation of platelets and endothelial cells and is partially released into the plasma in soluble form. It regulates platelet and leukocyte adherence to the active vascular wall and mediates part of the inflammatory effects in thrombosis [76]. P-selectin is important in thrombogenesis and induces a prothrombotic condition [45]. A soluble form of P-selectin exists in the blood circulation, either originating from an alternatively spliced form found in platelets and endothelial cells (lacking a transmembrane domain) or reflecting proteolytic cleavage from the cell membrane shortly after activation [77].

Soluble P-selectin (sPsel) has been characterized as a plasma marker of platelet activation as well as endothelial damage/dysfunction [78]. Endothelial damage and platelet dysfunction seem to be common in people who have suffered severe trauma. In a study on severely injured trauma patients, researchers observed that the levels of sPsel during the first week following trauma were slightly increased above the normal reference range, indicating ongoing activation of endothelial cells and platelets in these trauma patients. Despite the fact that this study focuses on the sPsel levels in trauma-induced coagulopathy, which is predicted to result in a reduced sPsel levels, the results demonstrated an increase in sPsel levels in trauma patients after one week of admission [79]. Hence, it can be concluded that sPsel levels may be employed as a biomarker for VTE in trauma patients.

P-selectin has become one of the novel biomarkers for VTE [80], considering its association with vascular and thrombotic diseases. P-selectin levels have been shown to be higher in patients with DVT in several studies [81,82]. Rectenwald et al. examined sPsel levels in 21 DVT patients and compared them with 30 healthy controls, finding that they were increased at 88.7 ng/mL (compared with 22.1 ng/mL) [82]. Antonopoulos et al. in 2014 conducted a meta-analysis comprising 11 trials (586 VTE patients and 1843 controls) and found that the pooled sensitivity for sPsel was 0.57, with a specificity of 0.73 [80]. In conclusion, sPsel may be the most promising emerging biomarkers for VTE, particularly in terms of offering a precise, noninvasive test that might rule out DVT without the need for further imaging.

Ramacciotti et al. examined 178 patients who were presented for duplex imaging to detect DVT using a logistic regression model. This study found that a combination of sPsel levels and Wells scores could establish the diagnosis of DVT with high specificity (96%) and a positive predictive value of 100%. As a result, sPsel levels greater than 90 ng/mL in combination with Wells scores of 2 or higher was found to be as effective as duplex ultrasound in ruling in the diagnosis of DVT, whereas sPsel levels less than 60 ng/mL in combination with Wells scores less than 2 was able to effectively “rule-out” the diagnosis [83]. In 2013, Vandy et al. employed the same criteria in a follow-up validation research of sPsel and found a specificity of 97.5% and a positive predictive value of 91% for the diagnosis of lower extremity DVT, with a sensitivity of 91% and a negative predictive value of 79% for its exclusion. They demonstrated that using a combination of levels of sPsel and D-dimer, and Wells score is a conclusive diagnosis to ruling in or ruling out lower extremity DVT in approximately one third of the patients [84].

The haemostasis and inflammatory biomarkers in predicting the risk of VTE have been summarized in Table 1 and Figure 3 below.

## 4. Imaging Techniques in Diagnosing VTE

Imaging techniques have rapidly evolved in the past few decades for VTE diagnosis. This includes computed tomography, pulmonary angiography, or a ventilation-perfusion scan to identify PE and duplex ultrasonography of the lower limb to detect DVT. These diagnostic procedures will be conducted based on clinical suspicion of VTE or for monitoring individuals who were contraindicated for VTE prophylaxis [85]

The current gold standard for diagnosing DVT is venous duplex ultrasonography, which combines color flow Doppler imaging with compression ultrasonography [86]. Duplex ultrasound imaging combines classical ultrasonography, which utilizes sound waves that bounce off blood vessels to make images, and Doppler imaging, which captures sound waves that reflect off moving things, such as blood, to assess their speed and other flow elements [87]. The absence of a complete compressibility of a venous segment under mild pressure is the main criteria for the identification of a first DVT incident [88]. In a recent meta-analysis study of DVT in the lower extremity, they found that the proximal leg, whole leg, and serial compression ultrasonographies had sensitivities and specificities of 90.1% and 98.5%, 94.0% and 97.3%, and 97.9% and 99.8%, respectively [89].

Compression ultrasonography may also be used to diagnose PE since a confirmed diagnosis of proximal DVT in persons with suspected PE is a significant predictor of PE and requires anticoagulant therapy. Importantly, a negative result does not rule out PE and needs further investigation, as revealed by a meta-analysis in which the sensitivities and specificities of proximal and whole leg ultrasonographies were 41% and 96%, and 79% and 84%, respectively [90]. However, according to a recent study, the use of selective duplex ultrasound in trauma patients was related to significantly reduced in-hospital PE as opposed to those who did not undergo routine ultrasound [91].

Lower extremity duplex ultrasound screening (LEDUS) has been proven to be effective in detecting asymptomatic DVT in trauma patients. An updated LEDUS technique for VTE screening was performed with a risk assessment profile (RAP) score ≥ 8 within 48 h of hospitalization in trauma patients. The earlier LEDUS may aid in the identification of patients with asymptomatic DVT upon admission after a traumatic injury [92].

In a review article by Maufus et al. using the Medline database, they concluded that ultrasound echogenicity or Doppler venous flow signal detection could not be utilized to diagnose recurrent DVT in lower limb patients. However, a new non-compressible vein or an increase in the diameter of a previously thrombosed venous segment by more than 4 mm is sufficient to validate the diagnosis of DVT recurrence, while an increase in diameter of less than 2 mm could eliminate the risk of DVT recurrence [93].

## 5. Conclusions

In conclusion, prolonged immobilization of trauma patients can induce a hypercoagulability state, resulting in the development of VTE. Routinely, only D-dimer is utilized as a screening test to determine the risk of VTE because of its high sensitivity. Due to its limitations in trauma cases, additional research is required to evaluate readily available biomarkers that show promising positive predictive value with high sensitivity and specificity in predicting the condition. In this present review, we summarize a few possible biomarkers that can be used to predict the risk of VTE in trauma care, which includes PT, aPTT, fibrinogen, thrombin generation, platelet count, anticoagulant activity of TFPI, CRP, ESR, and sPsel. Therefore, the addition of these new promising positive biomarkers to an existing risk assessment score could aid in early VTE preventive therapy in trauma patients without the need for imaging ultrasonography. Despite the availability of risk stratification and risk assessment tools for VTE, developing new risk assessment models that include these promising biomarkers could enhance the identification of patients who are likely to have VTE in a limited setting, thus allowing for targeted thromboprophylaxis for these patients.

## Figures and Tables

**Figure 1 diagnostics-13-00150-f001:**
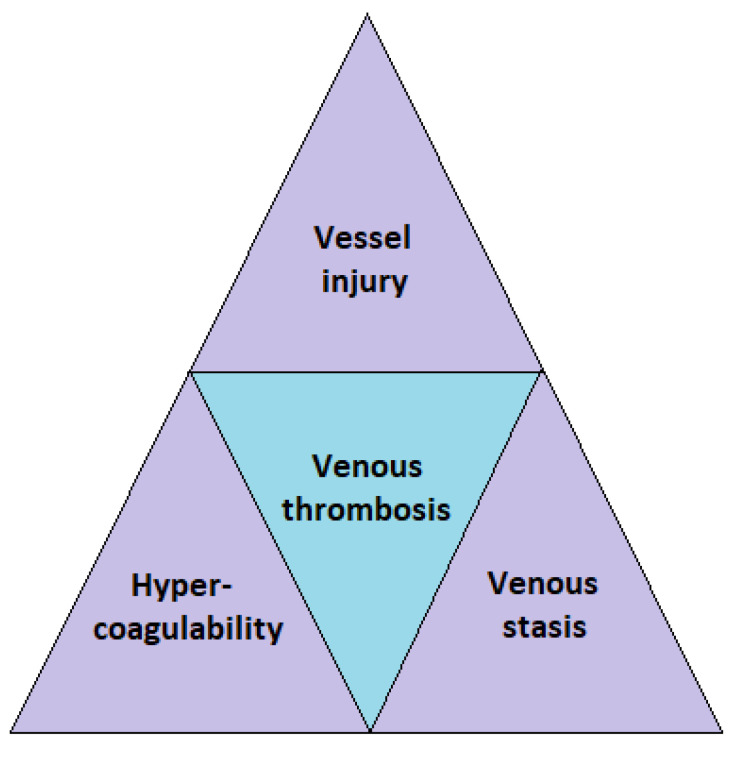
Virchow’s Triad.

**Figure 2 diagnostics-13-00150-f002:**
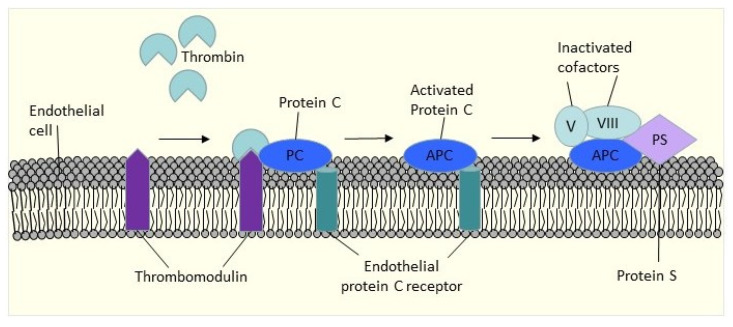
Protein C pathway.

**Figure 3 diagnostics-13-00150-f003:**
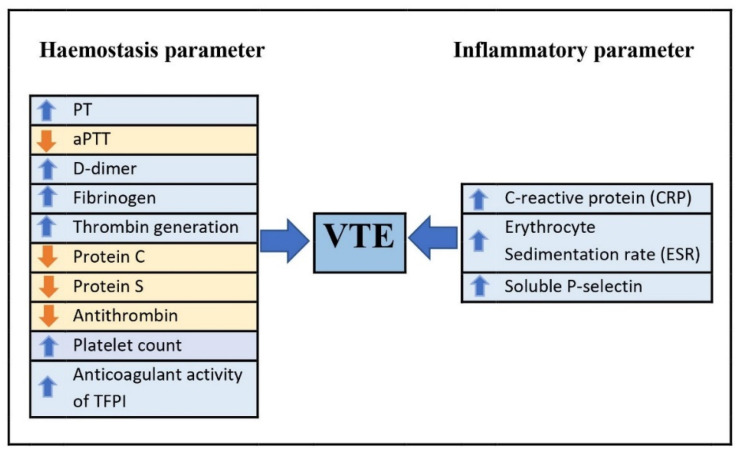
Haemostasis and inflammatory parameter in predicting VTE.

**Table 1 diagnostics-13-00150-t001:** Biomarker in predicting the risk of VTE.

Biomarkers	Results	References
PT	PT cutoff value more reliable in detecting coagulation factor deficiency with a sensitivity of 84%	[28]
	The PT optimal cut-off value for diagnosing DVT in trauma patients was 12.05 s with a sensitivity of 72.92% and specificity of 47.92% (AUC 0.617, 95% CI 0.505–0.730, *p* = 0.048)	[31]
aPTT	Lower aPTT was associated with a two-fold increased risk of VTE	[29]
D-dimer	The D-dimer optimal cut-off value was 3.825 mg/L with a sensitivity of 85.42% and specificity of 51.11% (AUC 0.624, 95% CI 0.505–0.744, *p* = 0.039) for diagnosing DVT in trauma patients	[31]
	D-dimer cut-off value of 4.01 mg/L for DVT in traumatic fracture patients, with AUC of 0.593, 71.2% sensitivity, and 44.8% specificity	[35]
Fibrinogen	Fibrinogen cut-off value of 3.543 g/L for the diagnosis of VTE in elderly fracture patients, with a sensitivity of 26.90% and specificity of 77.70% (AUC 0.5209, 95% CI 0.500 to 0.542)	[43]
Thrombin generation	First time to peak (ttpeak) value appears to be an independent predictor of VTE with a HR of 2.26, 95% CI [1.61, 3.18] per 1 (minute) decrease, *p* < 0.0001	[51]
Platelet count	Platelet count was one of the independent risk factors for DVT (*p* < 0.05), with a cutoff value of 200.5 × 10^9^/L, sensitivity of 38.9% and specificity of 85.9% (AUC = 0.642 (95% CI: 0.569~0.714))	[63]
TFPI	The optimal cut-off value for the TFPI initial anticoagulant time ratio for diagnosing DVT in trauma patients was 33.53%, with a sensitivity of 71.11% and specificity of 55.32% (AUC 0.646, 95% CI 0.533–0.759)	[31]
CRP	CRP was found to be higher in DVT patients (2.5 mg/dL ± 3.2) compared with controls (1.0 mg/dL ± 2.5)	[73]
ESR	An increase in ESR values was associated with the development of DVT before TKA	[75]
Soluble P-selectin	Increased levels of soluble P-selectin in DVT patients	[81,82]
	The pooled sensitivity of sPsel was 0.57 (95% CI = 0.30–0.82), while the pooled specificity was 0.73 (95% CI = 0.51–0.90).	[80]
	sPsel levels > 90 ng/mL with a Wells scores ≥ 2 was found to be effective in diagnosing DVT, whereas sPsel levels < 60 ng/mL with Wells scores < 2 was able to effectively ‘rule-out’ DVT	[83]

## Data Availability

Not applicable.

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
