# Peer review of "Haemostasis and Inflammatory Parameters as Potential Diagnostic Biomarkers for VTE in Trauma-Immobilized Patients"

_diagnostics, 2023, doi:10.3390/diagnostics13010150_

Round 1

Reviewer 1 Report

This is a very interesting and comprehensive narrative review that discusses possible biomarkers that can be used for predicting of VTE development in trauma patients. The review is well written and provides a lot of useful information on what is currently known about the subject.

My only suggestion is to add a section about imaging biomarkers. Duplex ultrasound is used as a screening test in trauma patients. While performing duplex ultrasound some markers can be registered such as flow velocity, spontaneous hyperechogenesity, etc. It would make a review more interesting if the information on the imaging biomarkers would be included.

Reviewer 2 Report

The authors analyzed in this review several biomarkers associated to the risk of developing DVT in patients with trauma. The paper is well written and exhaustive. Only few suggestions for the authors:

-          A table summarizing main data of each biomarkers can be added making readers more confident with these data.

-          Since most cited biomarkers are not currently evaluated in clinic management of patients with trauma, which one do the authors consider most promising to assess the risk of DVT and how would they modify antithrombotic prophylaxis according to them?
